# The Human-Centredness Metric: Early Assessment of the Quality of Human-Centred Design Activities

Olga Sankowski * and Dieter Krause 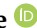

Hamburg University of Technology, Institute of Product Development and Mechanical Engineering Design, 21073 Hamburg, Germany; krause@tuhh.de
* Correspondence: o.sankowsk@tuhh.de

**Abstract:** Human-centred design as a research field is characterised by multidisciplinarity and a variety of many similar methods. Previous research attempted to classify existing methods into groups and categories, e.g., according to the degree of user involvement. The research question here is the following: How can human-centredness be measured and evaluated based on resulting product concepts? The goal of the paper is to present and apply a new metric—the Human-Centredness Metric (HCM)—for the early estimation of the quality of any human-centred activity based on the four goals of human-centred design. HCM was employed to evaluate 16 concepts, utilising a 4-point Likert scale, covering four different everyday products that were created by four students, which used three different human-centred design methods for this. The first concept was created without the application of any additional human-centred design method. The results illuminated trends regarding the impact of additional human-centred design methods on the HCM score. However, statistical significance remained elusive, potentially due to a series of limitations such as concept complexity, the small number of concepts, and the early developmental stage. The study's limitations underscore the need for refined items and expanded samples to better gauge the impact of human-centred methods on product development.

**Keywords:** design methodology; user-centred design; human-centred design; metric; experiment

## 1. Introduction

The concept of user- or human-centredness is gaining increasing importance in the field of product development. This is not only evident from the fact that it has fully entered product development thanks to various popular approaches, such as (lead) user innovation (e.g., [1,2]) and design thinking (e.g., [3,4]). In fact, the number of international standards dealing with the concept of human-centredness is also increasing; in 2002, the international standard ISO 9241: ergonomics of human–system interaction had only 17 parts, but now there are at least 40 [5]. The foundational concept of human-centredness entails prioritising the human or the user within a system, placing them at the core of deliberation [6]. As a result, this principle holds broad significance across various domains within both the social and corporate spheres. Within the existing rendition of the standard, this concept has consequently been expanded to encompass the notion of a "human-centred organisation", extending its reach beyond the confines of product development [7]. In the broadest sense, human-centred design is an approach to making interactive systems more user-friendly [8]. More precisely, a total of four human-centred quality goals are named in [8], which are to be achieved by using a human-centred design approach. These are usability, accessibility, user experience, and the avoidance of use-related damage.

However, this is where the common ground of methodological research around human-centred design ends. The research field is fed by a variety of research directions with findings from empirical and theoretical research. Among these are, e.g., human–machine interaction, cognition research, sociology, market research, and of course product development. Thus, new and partly transdisciplinary methods and techniques or variations of

existing ones are constantly emerging. The reason for this is assumed to be different and partly synonymously used terms, as well as different schools of thought [9,10].

Despite the international standards and their explanations and guidelines, the research field of human-centred design lacks conceptual foundations for methodological research. For this, we need a definite means of verifying that an activity within a product development process is indeed a human-centred activity. Only then is it possible not only to measure the achievement of certain objectives, but also to trace them back to human-centredness. However, the definition of a user-centred or human-centred activity remains ambiguous. What exactly does human-centredness mean and how can it be verified that an activity is indeed human-centred? Does this only pertain to activities that have been done with the involvement of end users, can all activities leading to a better human–system interaction be considered as human-centred, and what exactly are the boundaries of human-centred approaches? Kaulio [11] sorts various user involvement strategies into the categories Design with, Design by, and Design for. Sanders [12] has created a map of design practice and research in which she categorises different methods, approaches, and mindsets according to whether they consider the users of the system rather as subjects or as active co-creators. In [13–15], methods are classified according to the resources that they require, including the number of participants and their skills and knowledge. This implicitly underscores the necessity for some degree of user or customer engagement to transpire. In contrast, Overik Olsen and Welo [16] also explicitly list methods in their overview that are based on the use of their own intuition and statistical data, as well as the creation of scenarios. Their classification of human-centred activities is thus not only based on the involvement of users, but also includes activities and methods that they believe follow the same goal. Not only does this brief snippet of research make obvious the different perceptions of human-centred design, but there is no agreement on which activities belong to human-centred design and which do not [9]. Looking at the current research on the topic of the assessment and measurement of human-centredness or user-centredness, it becomes clear that the evaluation of usability and the associated aspects, such as ergonomics, satisfaction, or efficiency, is already part of the human-centred development process itself [17]. There are therefore many standard works and reference books on the subject of assessing the usability or user experience of products, e.g., Ref. [18]. These measurement methods are usually—in line with the human-centred mindset—designed for a certain level of user interaction and therefore require functional and physical prototypes. However, there are also efforts to implement the evaluation of human-centred aspects even further upstream in the development process. One possibility is to create virtual rooms or 3D scenes instead of physical prototypes, to place the products to be tested in them virtually and then have them evaluated by users. Bagassi et al. [19] evaluated the comfort of an aircraft interior design within a virtual reality setting. Virtual reality models were also used by Mengoni et al. [20] to measure users' emotional responses while interacting with household appliances; in another work, the functional accessibility of indoor objects was assessed within 3D scenes [21]. Similarly, Peruzzini et al. [22] evaluated human factors within a VR set-up of a workplace. For such measurements, new assessment methods are sometimes developed or adapted, e.g., Refs. [21–24]. In the search for measurement methods that can be used even earlier in the process, work can be found, for example, from the field of innovation research. For example, Genco et al. [25] measured the originality and quality of concepts created by undergraduate students after carrying out a creativity method. Zheng et al. [26] compared the effect of two concept selection tools on the quality and novelty of the concepts within a student team project. Haritaipan et al. [27] evaluated the benefits of magic-based inspiration tools for the concepts of novice designers in terms of various objectives, including creativity, excitement, and enjoyment. According to them, the evaluation was carried out with prospective users, but they also specified that these were exclusively people with an engineering design background.

The advantage of this type of measurement procedure and study design is that even a small change in the task for the participants—in this case, an additional design method—can

be measured and evaluated directly and immediately on the basis of its added value with regard to the achievement of the goal—in this case, an increase in innovation. Consequently, different design methods and their added value in terms of goal achievement can be compared with each other within an experiment. This is not possible yet within the context of human-centred methods and activities. Between the implementation of a method, e.g., a focus group, a prototyping workshop or a persona method, weeks and months can pass until the fulfilment of human-centred design goals, e.g., usability or user experience, can be proven with the help of established evaluation, e.g., with the use of prototypes. Many different factors apart from the methods may have influenced the design of the product in the meantime. No valid link can be derived between the methods and the product design, e.g., in the form of a measurement model, if a long time has passed between the input and the output to be measured. It is therefore also not possible to prove in such a setting that a specific human-centred activity had specific added value for the development of a user-friendly or accessible product. Therefore, and similar to innovation research, human-centredness needs a measurement method that allows for a very early and thus immediate evaluation of product concepts to close this gap. We define here that a maturity level of the concepts should be in the form of first-hand sketches and/or textual functional descriptions.

Such a measurement method would objectify the evidence for human-centredness independently of the possible schools of thought of the researchers and designers and allow the comparability and evaluability of human-centred techniques and activities with each other. The research question in this paper is the following: How can human-centredness be measured and evaluated based on resulting product concepts?

In this study, we aim to closely follow existing standards and develop a metric for the early assessment of product concepts based on the human-centred quality goals defined here. Such a metric allows us, on the one hand, to draw conclusions on the applied human-centred design methods and, on the other hand, to give insights into the potential for improvement in the product concepts. Thus, the Human-Centredness Metric (HCM) is developed and initially presented here.

In order to move closer to this objective, we will first provide a conceptual sharpening of human-centred design in Section 2.1. Based on this definition, we can subsequently derive a metric for the measurement of human-centredness and plan the experiment accordingly (Section 2.2). As this paper represents the first derivation and application of the HCM, our primary goal is to verify whether the proposed metric fulfils its requirements, whether it is applicable for the designated purpose, and whether it leads to plausible results. We further evaluate whether it leads to objective, reliable, and significant results. However, a complete validation assessment is not yet pursued at this stage. We explain our approach for validation and the experimental design in Section 2.3. Section 3 shows the results of this experiment and Section 4 discusses to what extent human-centredness is measured depending on the methods used. We conclude the paper in Section 5.

## 2. Materials and Methods

### 2.1. Defining Human-Centredness in Context of Product Development

In methodology research, forming classes and clusters of methods and approaches, as well as comparing and contrasting them, is a fundamental prerequisite for gaining insights. As shown in the Introduction, most authors define and distinguish between human-centred and non-human-centred methods depending on the user–designer interaction or whether some kind of user involvement is part of the method. However, this definition is difficult in exploring the added value that human-centred techniques have for product development. It is not possible to draw any clear conclusions about the extent of user involvement in the product design process on the basis of the naming of a method. Creativity methods such as paper prototyping or brainstorming, for example, can take place with strong user involvement or without any users at all. Are these particularly human-centred or not? There is also probably a difference in the quality of human-centredness depending on whether the product designers themselves have direct contact with end users or are merely presented

with processed data from an interview study by an external consultancy firm. Nevertheless, in both cases, data are collected from users. On the other hand, simulations of operation situations without any involvement of real users can also provide valuable insights and improvements in terms of usability. Since we wish to measure human-centredness as a quality feature of a product design, defining it on the basis of user involvement is not practical.

The ISO standards [8,17], whose basic descriptions we follow, define human-centredness, on the one hand, on the basis of the four quality objectives: usability, accessibility, user experience, and the avoidance of use-related damage. These are better suited to our objective but would themselves need to be defined more precisely and distinguished from other synonyms, such as applicability and usefulness. On the other hand, the general human-centred development process [17] is also described here. It is characterised by the fact that the context of use should be understood and defined prior to the definition of the requirements. All further human-centred activities are based on the context of use. The context of use is an essential concept because it can be used to fully describe the context of every user–product interaction. It includes the user, his/her goals and the tasks that he/she pursues through the product interaction, the resources and equipment necessary, as well as the environment and situation of the interaction [17]. The context of use is therefore well suited to deriving a more wide-ranging definition that can be used to unambiguously identify human-centred activities as such.

Figure 1 summarises the complete derivation of the definition for human-centred design. This will serve as the basis for deriving the metric and planning the experimental design. Human-centredness thus represents an overarching concept that can be projected at all levels of goal and action statements. On a very fundamental level, human-centred design can be described by its goal of making "interactive systems more usable by focusing on the use of the system" [8] and can be understood, according to an early definition by Abras et al. [28], both as an overarching mindset and philosophy and as a specific design approach itself. Within the more specific scope of design research, we maintain this distinction at two different levels of abstraction and do not make any significant deviations in the definition at the level of the mindset (Figure 1, left). On the right-hand side, however, in the definition of human-centred design as a design approach consisting of methods, tools, and techniques, we specify that any endeavour in the context of a product development process that addresses the collection of data or information about the context of use directly or indirectly from, with, or by the prospective users of the product is considered as a human-centred design activity. We emphasise the focus on the context of use through a symbolic image of the general human-centred design process according to [17] (Figure 1, right). For example, an observational study can provide information on the real and specific situation and environment as well as existing resources of a product interaction, whereas a questionnaire survey can only provide this information in a general and retrospective form but can also provide information on the goals of the product interaction as well as on the users themselves, e.g., gender, age, profession. A usability test with potential users or even the simulation of operation without user involvement tells a lot about the actual and concrete tasks that are necessary to achieve a usage goal. However, the two methods differ in the validity of their results. All these information elements resulting from the implementation of the methods are elements of the context of use.

This definition is a deliberate distinction from the understanding of some other researchers (e.g., Refs. [13–15]) in this field. It purposely includes activities that take place without direct user interaction but with a human-centred mindset. As a conclusion for this paper and the experiment planned therein, this definition has an important practical implication. As we are not reduced to methods with direct user involvement, we can choose rather simple and manageable empathy-based methods and design guidelines for the initial testing of the HCM. These types of human-centred methods are also particularly favourable for use in a controlled experimental environment, as they exclude possible biases in the test environment due to the involvement of external users.

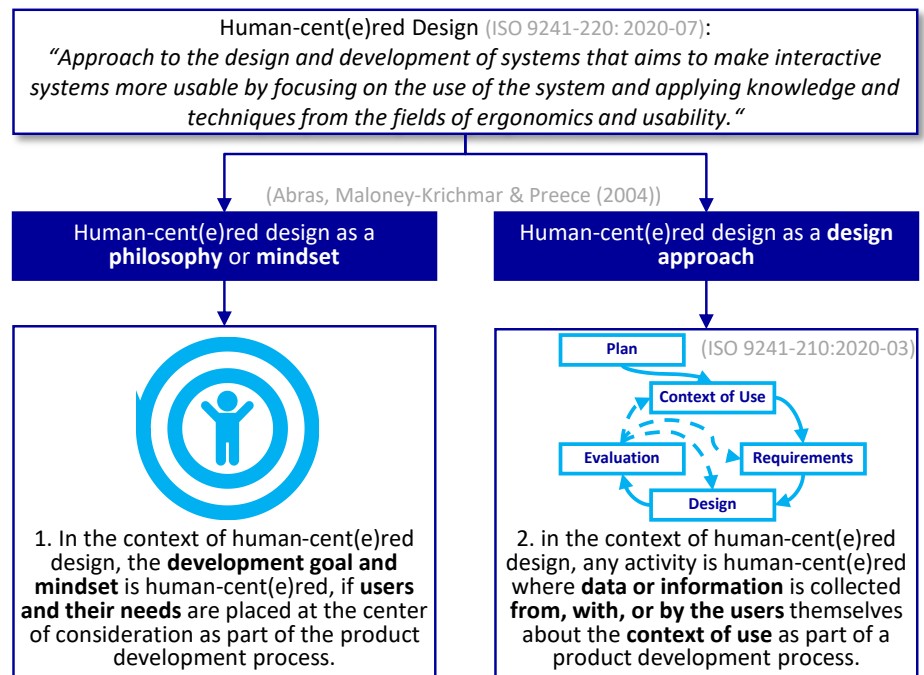

**Figure 1.** Deriving a working definition [8,17,28].

## 2.2. Evaluating Product Concepts

Before introducing our approach to measuring human-centredness, a brief explanation of the general types of methods that can be used to evaluate product concepts shall be presented here. Figure 2 illustrates classes of evaluation methods categorised along the dimensions "analytical vs. empirical" and "formative vs. summative". In this context, "empirical" refers to the evaluation of product concepts by potential users, as opposed to analytical evaluation, wherein evaluations are performed by experts, i.e., researchers or designers. Although it is theoretically possible to conduct all analytical evaluation methods of Figure 2 also together with potential end users, this requires certain technical and procedural know-how on the side of the participants in the method, which is usually either not widely distributed among the user group or would lead to a major restriction in the selection of the participants. The latter, however, also jeopardises the representative selection of participants of user involvement and cannot be recommended without restrictions. Haritaipan et al. [27] attempted something similar in their experiment, but included also only people with an engineering design background.

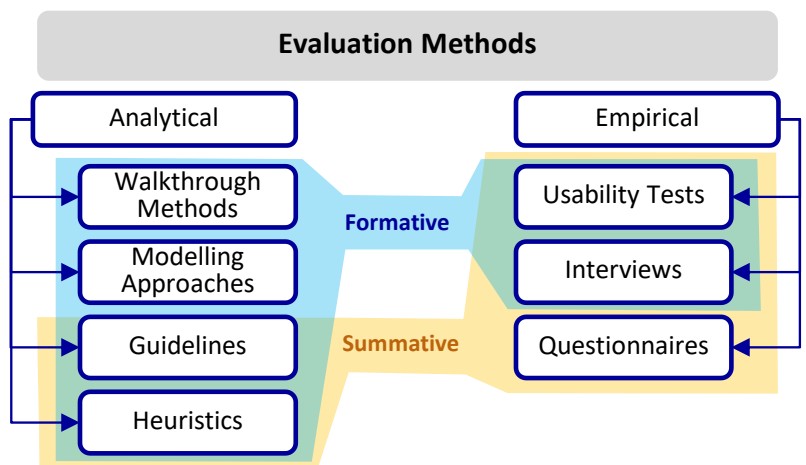

**Figure 2.** Classes of evaluation methods and differentiating dimensions.

Formative and summative methods differ in the point in the development process at which the evaluation can be carried out. However, as in the case of the distinction between analytical and empirical, this boundary is not a sharp line and has a certain intersection. Summative methods form a final judgment on an (almost) finished product. Without elaborated prototypes or established products on the market with a larger user community, summative approaches can hardly be performed. Formative procedures, on the other hand, allow the measurement of goal achievement in the ongoing development process at different stages of design. For example, interviews and different variations of them can be integrated at different points in the development process. They could be used in the form of the thinking aloud technique for a more formative evaluation already during development. For the purpose of a summative evaluation, interviews could be used in the form of asking users about past product interactions, as well as about any operating problems they encountered or their satisfaction with the product in retrospect. Interviews, as a class of methods, can therefore be assigned to both formative and summative approaches, but are always to be conducted empirically (Figure 2).

In alignment with our research inquiry, our objective is to establish quantifiable metrics for human-centredness that can be applied on the resulting product concepts. This approach aims to provide an immediate assessment, shortly after a human-centred method has been implemented, rather than relying on the documentation of the activities that have led to these product concepts over a longer period of time. Thus, we need a formative evaluation method that can be applied to the very early stages of development, long before physical prototypes and preferably even before detailed 3D models exist. Although there are formative approaches on the side of empirical methods (Figure 2), they still need at least basic functional prototypes for execution. We place our focus on the search for analytical methods that do not depend on the involvement of users, but still provide information about the context of use. Moreover, the metric that we derive should be applicable both to researchers specialising in human-centredness and to product designers themselves, facilitating prompt and early assessments of the efficacy of human-centred methodologies, as well as the identification of any vulnerabilities within the product concept. This means that the assessment, on the other hand, may also demand a certain level of know-how on the side of the method user and can be expert-based. Referring back to Figure 2, all classes of methods that are empirical and primarily summative can be excluded from the search for suitable evaluation methods. Apart from this, no further restrictions regarding the search for evaluation methods are necessary at the moment. Below, we present a concise overview of the evaluation methods already existing that meet the criteria of being formative and analytical. These methods have been categorised according to the four primary human-centred quality objectives as delineated in [8]: usability, accessibility, user experience, and the avoidance of use-related damage.

### 2.2.1. Evaluation of Usability

Usability includes the effectiveness, efficiency, and satisfaction of an interaction [29]. Among the four quality goals, it is probably the most important, since it forms the basis for the other goals. Without a minimum level of usability, it is difficult to have a good user experience; a product that is not well usable for a group of users will also have lower overall accessibility, and, finally, usability can exclude incorrect operation in parts and thus also use-related damage to a certain extent.

Most commonly used among the summative expert-based usability evaluations are heuristics. These are based on a collection of common usability problems or best practices related to usable design. Summarised as principles or even, more specifically, as guidelines, they can be used to identify common usability problems. The distinction between heuristic principles and (heuristic) guidelines is not sharp. Established usability principles can be found in [29]. Further guidelines and heuristics can be found, e.g., in [30–32]. Guidelines suitable for general product design and not specifically tailored to human-computer interaction can also be found in [33,34]. Walkthrough methods also work with criteria, but, in

contrast to heuristic procedures, here, the functions of the product are tested against the criteria by proxy for a specific user group over a time sequence, e.g., [35,36]. Among the analytical–formative evaluation methods, there are further the model-based approaches, such as GOMS [37] and its adaptations; it is supposed to map the user's mental models and predict his/her behaviour. Thus, it is an approach for process mapping from the users' point of view.

### 2.2.2. Evaluation of User Experience

User experience includes preferences, attitudes, and emotions, as well as the physical and psychological reactions, occurring before, after, and during use [8]. A basic model for describing user experience was developed by Hassenzahl [38,39]. The model divides the user experience into the pragmatic and hedonistic attributes of a product based on the product features. The latter is composed of stimulation, identification, and evocation and addresses the human needs for curiosity and pride. However, the model has only been used so far for summative user questionnaires, e.g., Refs. [38–40]. Similar to this, the User Experience Questionnaire (UEQ) from [41] is also constructed from adjective poles, which, however, can be classified differently to Hassenzahl into the two categories "perceived attractiveness" and "quality of the product".

As an extension to [38,39], Thüring and Mahlke [42] divided the user experience into three areas. These are the perception of instrumental qualities, the perception of non-instrumental qualities, and the emotional user reactions to the interaction with a system. The perception of instrumental qualities includes usefulness and usability, which we have already listed as separate goals. The remaining ones are the emotional user reactions, which include, e.g., subjective feelings and motor expressions, and the perception of non-instrumental qualities, which refers, among others, to aesthetic aspects and motivational aspects [42].

It is obvious that especially the strongly subjective emotional or hedonistic aspects can only be estimated with great uncertainty by means of expert-based evaluation or criteria lists. Consequently, there are only a few studies with analytical approaches. A checklist for avoiding negative user experiences based on best practices is available in [43]. Building on the model of [41], Mahlke [44] argues that the user experience can also be expressed in simplified terms by instrumental and non-instrumental aspects, since these trigger the emotional reactions in the first place. If usability aspects, i.e., the instrumental aspects, are excluded from the user experience, only the aesthetic aspects, symbolic aspects, and motivational aspects remain. Within these, visual aesthetics in particular can still be evaluated most reliably on an expert basis. Appropriately, Roussos and Dentsoras [45] present four main and 23 sub-criteria for the measurement of visual aesthetics.

### 2.2.3. Evaluation of Accessibility

Accessibility means the extent to which a product can be used by persons with the widest possible range of needs, characteristics, and abilities [8]. Design exclusion, on the other hand, is a measure of how many people are excluded from using the product due to lack of accessibility. For this, an exclusion calculator is available online [46]. Here, sorted by basic human abilities, requirements are defined that the interaction with a certain product demands from the user [47]. After entering product specific requirements, the percentage of the population excluded from using the product can be calculated on the basis of statistical population data. The criteria and their gradations, such as walking, concentration, or long-term memory, are not directly suitable for translation into evaluation criteria.

One further fundamental work in evaluating accessibility is the Principles of Universal Design [48]. These guidelines have been directly integrated, or sometimes adapted, into numerous national and international standards, such as those found in [49–52]. Their phrasing lends itself to more seamless translation into evaluation criteria.

### 2.2.4. Evaluation of the Avoidance of Use-Related Damage

The last objective, avoidance of use-related damage, is about preventing possible harm to human health, the economic status of stakeholders, and the environment [8]. In research, there are hardly any contributions that deal intensively and specifically with use-related damage or machine safety. Those that exist focus mostly on measuring and improving ergonomics to maintain work performance. However, this is already covered by the usability criteria.

Munshi et al. [53] examined the inter-rater reliability of machine safety scorecards extracted from standards and industry best practices and found them to be quite valid. Another study [54] used lost work days as an indirect metric for the safety or risk of facilities. Similar to the walkthrough methods, a failure modes and effects analysis (FMEA) [55] can also be performed for risk assessment. Finally, concrete evaluation criteria can be found in the machinery directive [56].

### 2.2.5. Complete Human-Centredness Metric (HCM)

In accordance with our objective, we primarily need lists of criteria from which a cumulative score for the evaluation of human-centredness can be derived. In the case of empathic methods as well as walkthrough methods, or process-based methods in general, which also include model-based approaches, this is more difficult to implement. For the derivation of a metric for the early measurement of human-centred goal setting, we therefore use the guidelines presented above and, if possible, give priority to those whose phrasing can be used for general product design as well as for early development phases.

In summary, we derive 21 criteria for the usability aspect, primarily from [33,34]. Based on the best practices from [43] as well as the criteria for evaluating visual aesthetics from [45], we obtain another 12 criteria for user experience. However, we do not list those criteria from [43] that are covered by the usability criteria, as well as those from [43,45] that are not suitable to be used in the early development phases, such as the evaluation of the colour style. For accessibility, we derive a total of eight criteria from [50–52]. For the avoidance of use-related damage, we obtain 13 criteria that can be anticipated at early development from the machinery directive [56].

Usability (21)

1. Interaction elements are simple and clear
2. Minimal number of actions required to complete task
3. Operations are effective, i.e., require low effort
4. Operations are efficient, i.e., require little time
5. Interaction elements are easily reachable when needed
6. Direct feedback about system status and operations
7. Intuitive perceptibility of functions and interaction elements
8. Intuitive understandability of functions and interaction elements
9. Helpful information is available when needed
10. Users have control over all functions
11. Users have the feeling of direct manipulation
12. Functions are easy to control
13. Users can correct errors when they occur
14. Design prevents errors by users to a large extent
15. Each interaction element is unique in terms of its use
16. Effort required to learn the operation is low
17. Operation is easy to memorise
18. Operation can be adapted to the needs of the user
19. Operation is internally consistent, i.e., design for elements with similar function is also similar
20. Operation is familiar from other areas to users
21. Operation and interaction elements are predictable for the user

User Experience (12)

1. Users do not get the feeling of incompetence in operation
2. Continuous work is possible/no forced interruptions
3. Operation creates a feeling of security/prevents feeling of insecurity
4. Design fulfils expected fundamental functions
5. Design has recognisable technological advance
6. Product form fits the context of use
7. Total number of visible individual parts is rather small
8. The system is simple and small, redundant functions are reduced as much as possible
9. Elements and functions are visually merged or organised into sets
10. Form and shape complexity, i.e., the total number of lines and curves, the change in their direction, and the amount and type of combinations of shapes that occur, is small
11. Proportions follow a harmonic design; dimensions differ for purely ergonomic and functional aspects
12. Product form follows a cohesive style characterised by unity, contrast, or balance

Accessibility (8)

1. Interaction elements can be detected by means of two sensory capabilities
2. Operating target can be reached via two independent paths
3. Interaction elements are intuitively recognisable
4. Elements and functions are clearly and noticeably labelled
5. Interaction elements are accessible with reduced mobility
6. Interaction elements can be operated with little physical strength
7. Working speed is reasonable and/or can be adjusted to the user
8. System uses a language that is easy to understand

Avoidance of Use-Related Damage (13)

1. The shape does not lead to mechanical hazards
2. Protruding elements do not lead to mechanical hazards
3. Moving parts do not lead to mechanical hazards
4. Moving masses do not lead to mechanical hazards
5. Elastic elements do not lead to mechanical hazards
6. Unhealthy postures are avoided
7. Accidental activation of the system does not lead to hazards
8. Failure of the system does not lead to hazards
9. There is no danger to people if safety devices are bypassed
10. Accidental movements of people (e.g., slipping, tripping, falling) do not lead to hazards
11. Insufficient information does not lead to hazards
12. Movement of people around the facility does not lead to hazards
13. Emergency stop function is provided/absence of emergency stop does not lead to hazards

### 2.3. Experimental Design

With reference to the overall aim of the experiment presented here and its design, we loosely follow the descriptions, models, and references of Blessing and Chakrabarti [57], Pedersen et al. [58], and Üreten et al. [59,60]. These contain descriptions of procedures for conducting validation studies in the context of design method research, as well as the levels of validity of a design method that should be evaluated over the course of various studies. Although the HCM presented here is not a design method per se, but rather a tool for analysing systems, the models from the field of design method research appear to be helpful for our research. For example, Blessing and Chakrabarti [57] describe the first stage of the validation of a method as "support evaluation" in their framework for the design

research methodology. It provides initial evidence that the method developed meets the requirements and is internally consistent, complete, and applicable. They do not narrow down exactly how the proof is to be provided. Pedersen et al. [58] call this stage of validity "empirical structural validity"; they describe that this step serves to prove the correctness of the method and the appropriateness of the example application. Thus, the method fulfils the intended function within the setting of an appropriate example. They recommend that this should be demonstrated through an empirical study, but in a qualitative way. The work of Üreten et al. contains practical models for the step-by-step implementation of empirical studies to test the applicability and usefulness of design methods [59], as well as best practices regarding the implementation of empirical studies with students [60]. In this first experiment, we aim at proving the consistency and applicability of the HCM to realistic and appropriate sample applications (support evaluation/structural validity) and, at least not at this point, less at proving usefulness. The insights from [60] were mainly used for the planning of the experiment design, whereas [59] added some more dimensions to this.

Thus, a study design was created that allows a minimum of comparable concepts for the application of the HCM despite the limited resources available. Framework of implementation and curricular integration. As our research on this topic took place outside of a funded project and therefore no financial compensation for student participants could be paid, only the integration of the experiment within a lecture was possible. Participant level. A suitable course was found in the undergraduate programme. Restrictions with regard to learning objectives. In the course in question, students are supposed to write a scientific report on a technical topic. This learning objective allowed us great freedom in the design of the experiments and did not lead to any restrictions in terms of content. Number of participants. Due to the effort and lack of research funding, only a very small number of participants could be involved in the experiment. Length of study, number of sessions, and length of sessions. However, it was advantageous that each student could participate in several sessions due to the duration of the course (one semester) and that the sessions themselves did not have strict time constraints; they did not last longer than half a day each. It was therefore decided that four undergraduate students from comparable engineering courses (mechanical engineering and general engineering) would each produce a total of four different concepts to which the HCM evaluation would be applied afterwards. Students wrote their reports about the human-centred design methods that they used and also wrote a summary of the design concepts that they created. The quality of the concepts was not graded within the context of the teaching unit. However, the participants were informed that these concepts were to be evaluated as part of a series of experiments separate from the teaching unit.

Expertise of participants. Although all four participating students were in higher stages of their studies and had also attended several basic lectures on the conception and design of technical systems, their knowledge of design methods or even human-centred methods was comparably low. Type of task. Within the tasks of each session, the participants were asked to devise a new concept in the form of sketches for a technical device. For this, they were also required to produce several intermediate results within the framework of a general design approach. Furthermore, they had to address human-centred design goals with the help of another additional method or tool. In order to avoid the influence of individual preparation, the students were neither informed about the development goal nor about additional human-centred tools before the start of a session. However, the material was prepared in such a way that the participants had sufficient time to both understand and apply the complementary human-centred tools during the session. Four different products from everyday life were selected as the focal point for development in order to compensate for the participants' limited expertise in specialized technical systems.These products include a washing machine, a vacuum cleaner, a coffee machine, and a Blu-Ray player. These were chosen to be as diverse as possible in terms of their potential challenges in operation; for example, a washing machine is more likely to

pose challenges to the user in terms of physical strength and accessibility, and a Blu-Ray player is more likely to pose challenges in terms of cognitive understanding.

Variation of parameters. Due to the small number of participants and the multiple runs, it was decided to diversify the parameters of the experimental set-up as much as possible. Therefore, all four participants had to create a concept for each of the four products, with alternating order of the sessions. The methodological support was also varied in each case, so that there was no combination of development goals, sequences of development, and additional methodological support used twice within the set-up. There was only one constant in the sequence of sessions. Identical in all four runs for each student participant was that the first experiment was conducted without an additional human-centred design method (M0). This served to calibrate the participants with regard to the general development process and was intended to avoid any possible overload in the first session. Figure 3 shows the overview of these variations. For example, participant 1 created a new washing machine in the first session without any additional human-centred method or tools (M0), while participant 2 had this task only in the last session and was also supposed to use an additional human-centred tool (M0 + M1). This set-up was expected to result in the widest possible range of appropriate examples for the initial evaluation of the HCM, even if this was at the expense of validity in terms of demonstrating the effectiveness of the human-centred techniques and the reproducibility of the results.

| Student participant | Experiment 1 | Experiment 2 | Experiment 3 | Experiment 4 |
|---|---|---|---|---|
| 1: female undergrad student, Gen. Eng. | **M0** | **M0+M1** | **M0+M2** | **M0+M3** |
| 2: male undergrad student, Mech. Eng. | **M0** | **M0+M2** | **M0+M3** | **M0+M1** |
| 3: male undergrad student, Gen. Eng. | **M0** | **M0+M3** | **M0+M1** | **M0+M2** |
| 4: male undergrad student, Mech. Eng. | **M0** | **M0+M1** | **M0+M2** | **M0+M3** |

M0: General approach to design  M2: Persona-scenario method
M1: Universal Design principles  M3: Dialogue principles

**Figure 3.** Tasks and procedure of the experiments.

Support. The participants worked on the tasks alone but under supervision and were allowed to use a PC without internet access for documentation purposes. There was no assistance in the technical design or the interim results from the experimental supervisor; only questions regarding the procedure were answered. Documentation. Apart from the results that were produced directly in the sessions, the students also wrote summaries of the concepts as part of their reports.

Task and experimental procedure. In preparation for the experiments, students were required to repeat the general approach to design [61]. This is an established and fundamental approach to methodical and structured product design that the students already knew from their previous studies. The general approach to design [61] was given in all method

experiments to ensure a minimum level of structured work and served here as a calibration method (M0). As part of the approach, participants always had to write requirement lists and function structures and build morphological matrices, additionally to creating concepts for the task given.

Disturbance variables and other variables. The students had up to four hours to work on each of the assignments and were allowed to take breaks at their own convenience. Each assignment consisted of a task description including an objective, hints, and pictures of three sample products. There was no additional information about the sample products given to the students, and no technical drawings or function descriptions. Except for the objective and the sample products, the task descriptions were identically phrased for each assignment, regardless of the product to be designed. For example, the objective in the case of the coffee machine assignment was "Find new, innovative, and user-friendly concepts for a coffee machine resp. the main function of making coffee". The objectives for the other products were formulated analogously. In addition, the students were advised in the hints that they should focus primarily on parts and functions in the product concept that were in direct interaction with the operator or were noticed by him/her. This experimental design and task setting were intended to keep disturbance variables and other variables under control as far as possible.

Design of the independent variable. The goal of a controlled study environment was found to be challenging with regard to the selection of human-centred design methods. In the case of most approaches, we found it difficult to conduct them under the same boundary conditions or in the form of a simulation. Interviews or focus groups are difficult to simulate without user interaction, but a real or simulated user interaction seemed to be a major disturbance variable. The presentation of, e.g., simulated questionnaire data also appeared to us to move beyond the time frame of the study, as only half a day could be planned per session and the evaluation and interpretation of user data can be quite time-consuming, especially for inexperienced designers. Therefore, the human-centred approaches had to be selected or prepared in such a way that they could be conducted alone by the student, without direct end user interaction, and that they could be understood and applied within a few hours. In order to minimise learning effects in the course of the sessions, three different methodological approaches were chosen that were as different as possible in terms of their goals or approaches. The Principles of Universal Design (M1) [48], the Persona Scenario Method (M2) [62], and the Dialogue Principles (M3) [63] were selected here. Although these three are not equally mature design methods—in the case of the guidelines, they are rather tools—they are nevertheless applicable in the context of early product development and still met our further requirements by far the best. For each, a method profile was prepared. It consisted of a short summary of the basic idea and goals as well as the procedure of the method. In addition, the reference papers were provided.

The Universal Design Principles (M1) are a set of guidelines or best practices about designing products, buildings, and information in such a way that they can be used by as many different people as possible without restriction or the need for adaptation [48]. Universal Design is thus intended to increase the accessibility of products. There are a total of seven principles, such as Equitable Use, to which several guidelines are assigned. The principles can be integrated at various points in the development process.

In the Persona Scenario Method (M2), one or more personas undergo one or more interaction scenarios; as a result, new insights about the user and the interaction as well as design ideas will result [62]. This method thus can enhance the user experience for the user. Personas are fictitious characters with detailed descriptions of many different attributes, e.g., age, occupation, and consumption habits. The attributes are based on results from field research or on the empathy of the designers and should represent the users of the target system [64]. In order to adapt to the scope of the experiment, students were given a persona description. This allowed them to skip the first step and hand over directly to the interaction scenarios and gain insights from them.

The Dialogue Principles (M3) [63] can be applied in a similar way to the Universal Design Principles. There are also a total of seven principles that are aimed at improving the usability of a human–machine interaction (dialogue). Although the description and examples of the principles in the standard are partially directed towards interaction by means of a screen/display, the principles are universally translatable to any interaction between man and machine. Examples of this are conformity with user expectations and error tolerance. The hypothesis to be investigated was therefore as follows.

**Hypothesis (H0):** *The application of human-centred design methods will lead to a significant change in the human-centred quality goals of the developed concepts.*

The alternative was as follows.

**Hypothesis (H1):** *The application of human-centred design methods will not result in a significant change in the human-centred quality goals of the developed concepts.*

Since we also suspected a potential influence of other factors, besides the use of additional human-centred design methods, these were also examined. These other factors were as follows.

- The participants, i.e., that one participant develops significantly better concepts regarding the human-centred quality goals than the other participants.

**Hypothesis (H2):** *Participants with similar background knowledge will develop concepts with significant differences in terms of human-centred quality goals.*

- The order of development, i.e., those concepts that were developed last by the participants achieve higher values regardless of the methods used.

**Hypothesis (H3):** *Concepts developed at the end of the series will achieve significantly higher values regarding human-centred quality goals than concepts developed at the beginning of the series.*

- The product examples, i.e., that, for example, Blu-Ray players achieve higher values on average than, for example, washing machines, regardless of the methods used.

**Hypothesis (H4):** *The achievement of the human-centred quality goals depends significantly on the chosen product example.*

For the presentation and summary of the results, we use descriptive statistics, and, for the examination of the hypotheses, we compare the mean values. Due to the small sample (n = 16 concepts), we cannot assume a normal distribution. For the comparison of the human-centred design methods and the influence of the order of the experiments, we deal with dependent samples; here, we use the Friedman test [65], which is applicable for not normally distributed and dependent samples. When comparing the effects of the choice of product examples and the participants themselves, we deal with independent samples; here, we use the Kruskal–Wallis test [66], which is used for not normally distributed and independent samples.

Furthermore, we aim to improve the HCM itself. The experiment presented here, also due to the small size of the sample (n = 16 concepts), rather serves as an initial validation of the applicability and structure of the metric (i.e., support evaluation, according to [57], or structural validity, according to [58]). Therefore, further evaluation and statistical procedures will be used to improve the quality of the metric for future application. On the one hand, we look at which items particularly often cause difficulties in the assessment and should therefore be optimised in terms of their phrasing in future studies. Secondly,

the internal consistency of the items is examined with the help of Cronbach's alpha [67]. It indicates how well a group of variables or items measures a single, unidimensional latent construct. In other words, it is a measure for the correlation between the items in a metric. Finally, a factor analysis is carried out for the individual items. The four human-centred quality goals of usability, user experience, accessibility, and avoidance of use-related damage and their items partly overlap in their meaning. With the help of an explorative factor analysis, the items are detached from these four original quality objectives and combined into new factors that are meaningful and as independent from each other as possible. An exploratory factor analysis is better suited for the discovery of a new structure than a confirmatory one, which serves to reaffirm an existing structure.

## 3. Results

### 3.1. Results of Student Experiments

Aside from the sketches, students prepared requirement lists, morphological matrices, functional structures, and complementary descriptions of the concepts and their usage; these results were used in their entirety for the analysis via the HCM. Figure 4 shows an example of the type and quality of the results that emerged from the sessions. For further understanding of the paper, it is not necessary to fully comprehend the concepts and descriptions produced by the students; however, it becomes clear that these concepts, used for the application of the HCM, are indeed at a very rough stage of development.

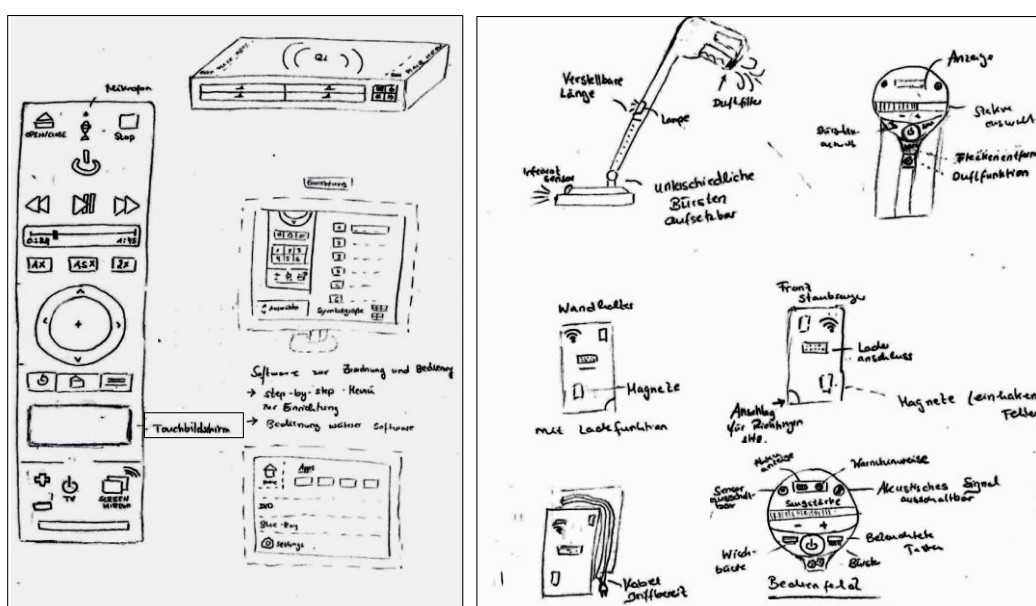

**Figure 4.** Examples of resulting concepts consisting of sketches and additional information.

### 3.2. Assessment of Concepts

The 16 concepts, consisting of requirement lists, functional structures, sketches, and descriptions, were evaluated with each item of the metric from Section 2.2 using a 4-point Likert scale. The response options were as follows: probably fulfilled, rather fulfilled, rather not fulfilled, and probably not fulfilled. If no statement could be made about the probable fulfilment of the item, "not applicable" was selected. In order to improve the comparability of the product concepts, a product group was always evaluated in one instance when evaluating the concepts. This means that all washing machines were evaluated first, then all vacuum cleaners, etc.

Figure 5 graphically shows the results for the HCM score for all 16 concepts sorted by the methods used, i.e., concepts created with the same methods are grouped together. The order within the blocks of four from left to right is always first washing machine (WM), then vacuum cleaner (VC), then Blu-Ray player (BRP), and finally coffee machine (CF). This

means that, for example, the second column in the fourth block of four, "+M3: Dialogue Principles", shows the results for the vacuum cleaner concept, which was developed with the help of the general approach to design (M0) and the Dialogue Principles (M3). The sorting therefore does not directly give any information about the participant who developed the concept or the order of development. For this, an examination of Figure 3 is necessary. Accordingly, in this example, the vacuum cleaner concept with the additional method M3 was developed by the fourth participant as the fourth concept in the row.

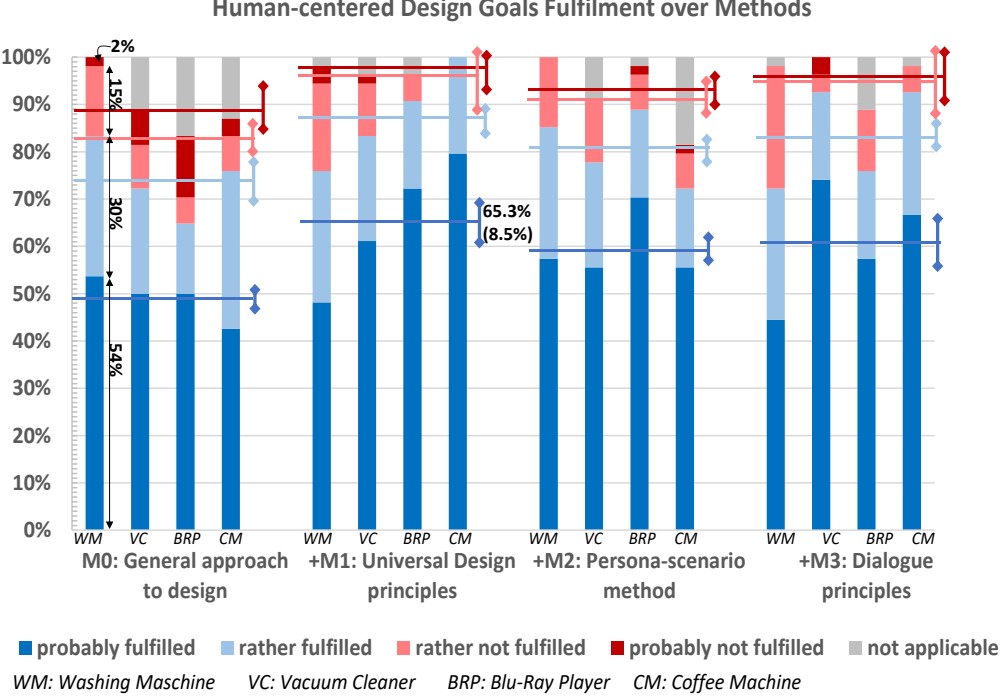

**Figure 5.** HCM scores depending on additional design method for each concept.

The individual results are shown as a stacked bar chart. This can be read as follows: approximately 54% of the items for the washing machine concept without an additional human-centred method (first column of the first block of four) were rated as probably fulfilled (dark blue); another approximately 30% as rather fulfilled (light blue); approximately 15% as rather not fulfilled (light red); approximately 2% as probably not fulfilled (dark red); and 0% as not applicable (grey). In addition, the mean values (horizontal lines) and the standard errors of the mean values (vertical lines) are plotted for each group of four. For example, the mean value of the probably fulfilled items of all four concepts for which the Universal Design Principles were used as an additional method (second group of four) is 65.3% and the standard error of this mean value is 8.5%.

Figure 5 provides a good visual representation of trends in the added value of human-centred approaches. Here, one can directly see that the concepts that emerged with additional methodological support had overall higher mean values with regard to the number of probably fulfilled items. That means that the dark blue horizontal for the three right groups of four (+M1, +M2, +M3) always higher than the dark blue horizontal for the first group of four on the left (M0). In the same way, the proportion of items that are probably not fulfilled is always smaller than on the left. That means that the distance between light red and dark red horizontals for the three right groups of four (+M1, +M2, +M3) is always smaller than the same distance for the first group of four on the left (M0). However, this representation also shows the sometimes very large standard errors and tells us nothing about whether the values are significant. It is also not possible to tell whether they differ due to the change in the independent variable—an additional human-centred approach—or whether the cause is rather individual differences between the participants or learning

effects. The mean values of the concepts are therefore further checked for significance with IBM® SPSS® Statistics, version: 28.0.1.1 (14).

A further representation of the results, extracted from SPSS®, can be seen in Figure 6; the associated descriptive statistics' values can be found in detail in Table 1. Here, the mean values of the metric per human-centred design goal are listed individually and as a sum value (M_HumanCentredness) sorted by the methods used. For this mean calculation, the Likert scale was converted into metric values, i.e., probably fulfilled = 4, rather fulfilled = 3, rather not fulfilled = 2, probably not fulfilled = 1. Individual items marked as not applicable were excluded from the mean calculation. The minimum value for the overall human-centredness score (M_HumanCentredness) was thus lowest for the concepts without the additional use of human-centred methods (M = 3.3, SD = 0.05) and highest for the concepts for which the Universal Design Principles (M1) were additionally used (M = 3.55, SD = 0.25); in the latter, however, the standard deviation was also higher.

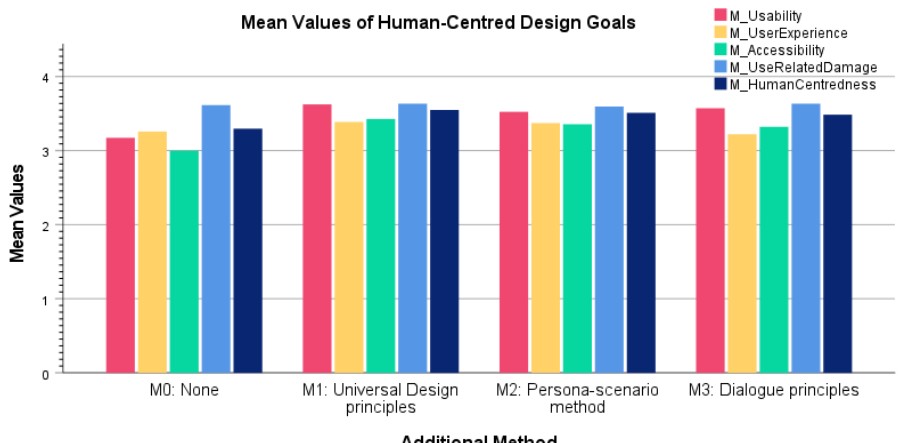

**Figure 6.** Mean values of human-centred design goals sorted by the additional methods used.

**Table 1.** Descriptive statistics on the fulfilment of human-centred design goals, sorted by the additional methods used.

| Additional Method | | N | Mean | | Std. Deviation | Variance |
|---|---|---|---|---|---|---|
| | | Statistic | Statistic | Std. Error | Statistic | Statistic |
| M0: None | M_Usability | 4 | 3.17 | 0.070 | 0.140 | 0.020 |
| | M_UserExperience | 4 | 3.26 | 0.121 | 0.242 | 0.059 |
| | M_Accessibility | 4 | 3.00 | 0.117 | 0.233 | 0.054 |
| | M_UseRelatedDamage | 4 | 3.62 | 0.160 | 0.320 | 0.103 |
| | M_HumanCentredness | 4 | 3.30 | 0.028 | 0.055 | 0.003 |
| M1: Universal Design Principles | M_Usability | 4 | 3.62 | 0.097 | 0.194 | 0.038 |
| | M_UserExperience | 4 | 3.39 | 0.199 | 0.399 | 0.159 |
| | M_Accessibility | 4 | 3.43 | 0.130 | 0.261 | 0.068 |
| | M_UseRelatedDamage | 4 | 3.63 | 0.187 | 0.374 | 0.140 |
| | M_HumanCentredness | 4 | 3.55 | 0.126 | 0.252 | 0.063 |
| M2: Persona Scenario Method | M_Usability | 4 | 3.52 | 0.126 | 0.251 | 0.063 |
| | M_UserExperience | 4 | 3.37 | 0.095 | 0.190 | 0.036 |
| | M_Accessibility | 4 | 3.36 | 0.180 | 0.360 | 0.129 |
| | M_UseRelatedDamage | 4 | 3.60 | 0.176 | 0.352 | 0.124 |
| | M_HumanCentredness | 4 | 3.51 | 0.040 | 0.079 | 0.006 |
| M3: Dialogue Principles | M_Usability | 4 | 3.57 | 0.101 | 0.203 | 0.041 |
| | M_UserExperience | 4 | 3.22 | 0.122 | 0.245 | 0.060 |
| | M_Accessibility | 4 | 3.32 | 0.090 | 0.178 | 0.032 |
| | M_UseRelatedDamage | 4 | 3.63 | 0.176 | 0.352 | 0.124 |
| | M_HumanCentredness | 4 | 3.49 | 0.103 | 0.206 | 0.043 |

In this evaluation, tendencies with regard to the differences between the mean values are no longer so clearly visible. The mean values for the usability items (M_Usability) still show the clearest differences (Figure 6, red bars); the mean value without additional methods (M0) is M = 3.17, while it is at least M = 3.52 for the concepts with additional methods (Table 1). The mean values for accessibility (M_Accessibility) also indicate a slight positive trend due to additional method use (Figure 6, green bars); the mean value without additional methods (M0) is M = 3.0, while, for the concepts with additional methods, it is M = 3.32 and higher (Table 1). The mean values for the avoidance of use-related damage (M_UseRelatedDamage), on the other hand, do not indicate any change due to different methods (Figure 6, light blue bars); all mean values are around M ≈ 3.6 (Table 1). Meanwhile, the mean values for user experience (M_UserExperience) (Figure 6, yellow bars) for both the concepts without the additional use of methods (M0) and for those with the additional use of the Dialogue Principles (M3) reach comparatively low values in the fulfilment of the items, M ≈ 3.2 (Table 1). Again, these descriptive statistics do not tell us anything about significance, but they do give us an indication of which items or groups of items are imprecise in their formulation or in the measured variables and therefore need to be optimised. Further analyses and measures to improve the items of the HCM will follow in Section 3.3; the question of significance will be examined in the following with the help of suitable sample tests.

Regarding Hypothesis H0, the Friedman test [65] is applied, which is suitable for small, non-normally distributed samples as well as for connected samples. Since Hypothesis H0 was derived directly from the research question, not only the mean value of fulfilment across all items (M_HumanCentredness) is compared here, but also the individual goals. As can be seen in Table 2, none of the tests fall below the significance value of $p$ = 0.05, whereby the mean values of the usability and accessibility goals still show the greatest differences between the individual methods. The test statistic for the user experience shows a rather low value, $X^2 \approx 2.6$, while, for the avoidance of use-related damage, it shows no deviation at all, $X^2 \approx 1.0$. The statistical test cannot prove any significant changes through changes in the methodological support. Hypothesis H1 cannot be rejected and H0 cannot be accepted on the basis of this study.

**Table 2.** Friedman test results on significance of fulfilment of human-centred design goals in dependence on the additional methods applied.

| Quality Goals | Total N | Test Statistic $X^2$ | Degree of Freedom df | Asymptotic Sig. $p$ |
|---|---|---|---|---|
| M_HumanCentredness | 4 | 3.900 | 3 | 0.272 |
| M_Usability | 4 | 6.474 | 3 | 0.091 |
| M_UserExperience | 4 | 2.605 | 3 | 0.457 |
| M_Accessibility | 4 | 4.500 | 3 | 0.212 |
| M_UseRelatedDamage | 4 | 1.000 | 3 | 0.801 |

In addition, the effects of the participants themselves, the order of the experiments, and the product samples on the sum of all items is examined. Since the participants and the product examples are independent samples, we use the Kruskal–Wallis test [66]; the significance threshold value is always $p$ = 0.05.

With regard to the student participants and their possible heterogeneity, there are no significant differences between them ($X^2$(3) = 1.346, $p$ = 0.752). Hypothesis H2 can thus be rejected.

With regard to the order in which the experiments were conducted, there were no significant differences between the concepts developed at the beginning and those developed at the end ($X^2$(3) = 5.400, $p$ = 0.145). For a more detailed investigation of possible learning effects, the achievement of the usability goal alone was also considered in relation to the

sequence. Here, the change from experiment 1 to experiment 3 fell below the threshold value ($X^2(3) = -2.375$, $p = 0.009$); the change from experiment 1 to experiment 4, however, again showed no significance ($X^2(3) = -0.875$, $p = 0.338$). Hypothesis H3 is thus also rejected.

In relation to the product examples used, again, no significant differences could be found with regard to the achievement of the human-centred design goals ($X^2(3) = 4.809$, $p = 0.109$). Hypothesis H4 is also rejected.

The calculation of the effect size due to lack of significance is waived.

### 3.3. Improving the Human-Centredness Metric (HCM)

Since the applicability, consistency, and structure of the HCM are also to be further improved, further statistical investigations are carried out for this purpose.

Firstly, an overview is given of those items whose application was not possible in every case and were therefore rated as "not applicable". As the diagram in Figure 7 shows, this occurred particularly with items from the areas of usability and user experience, whereby all but one of these items from the user experience category can be assigned to the visual aesthetics evaluation (User Experience No. 6–12). This indicates the need for the refinement of item phrasing, with particular attention to improving the user experience evaluation, especially in the area of aesthetics evaluation.

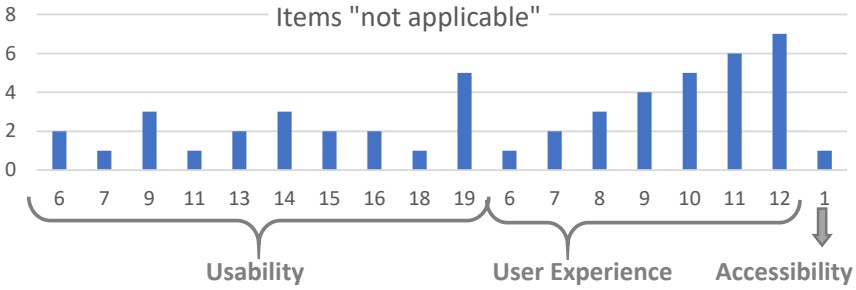

**Figure 7.** Frequency of items marked as "not applicable".

Likewise, another measure of the applicability of the metric is the number of items that achieved complete fulfilment in all 16 concepts, meaning that they had no effect on the total HCM score. This is the case for Usability No. 12, User Experience No. 4, Accessibility No. 7, and Avoidance of Use-Related Damage No. 1, 3, and 4. This also indicates that the items were formulated too imprecisely and thus were always marked as probably fulfilled.

To check and improve the internal consistency of the items, the value of Cronbach's alpha [67] is calculated. In all 16 cases and a total of 54 items, the value is a = 0.830, which can be interpreted as a "good" value. The detailed evaluation shows that this value can hardly be improved by omitting individual items. By omitting one of the items Usability No. 16, 18, or 21, User Experience No. 3, or Avoidance of Use Related Damage No. 5, 6, 8, or 9, a maximum alpha value of up to a = 0.835 could be achieved. This does not seem to be a large effect, but should be investigated again in future experiments with a larger sample.

Finally, the question remains as to whether the division of the HCM into more defined and non-overlapping factors is possible. The initial categorisation into four groups—usability, accessibility, user experience, and avoidance of use-related damage—proved valuable in developing the metric initially. However, this approach occasionally results in the dispersion of related topics, such as ensuring safe operation, across multiple categories.

This can be improved with the help of a factor analysis. To begin, the number of relevant factors must be determined. It depends on how much variance of the item correlation can be explained by the individual factors. The explained variance per factor corresponds to its eigenvalue, whereby all factors with an eigenvalue >1 are to be considered according to the Kaiser–Guttman criterion [68,69]. Based on this analysis, the 54 items should be allocated across 11 factors, collectively accounting for 93.6% of the variance in all variables. While the cumulative variance is notably substantial, the 11-factor allocation might appear relatively extensive. Therefore, we further wish to examine the scree plot.

According to the scree test [70], the eigenvalue plot is examined for a "kink" and the number of factors is selected whose eigenvalues lie above the kink. In our case, these are six factors (Figure 8, red horizontal line).

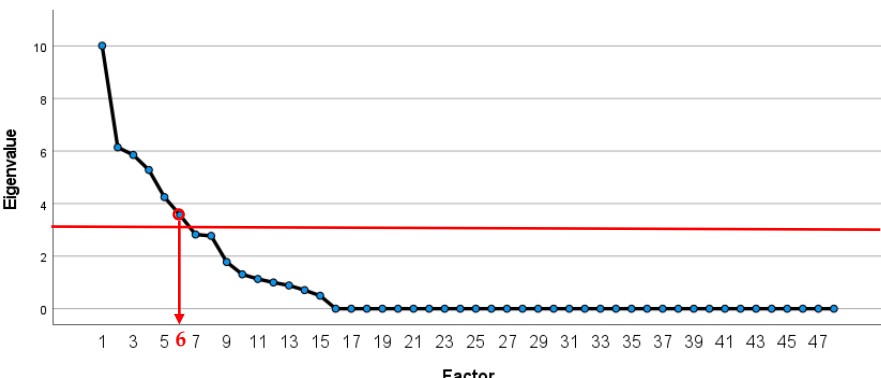

**Figure 8.** Scree plot.

We decide on six factors accordingly and sort the items that have the highest loadings for the six factors in each case. If an item has a similarly high loading for more than one factor, these are assigned to the most suitable factor in each case according to their meaning and not their loading. The same applies to the six items that cannot be evaluated in the factor analysis due to their 100% degree of fulfilment, i.e., Usability No. 12 (USA_12), User Experience No. 4 (UEX_4), Accessibility No. 7 (ACC_7), and Damage Avoidance No. 1, 3, and 4 (DAM_1, DAM_3, DAM_4). This results in the following six factors with their respective items. For better traceability, the abbreviations of the original sorting have been added for each item.

1. **Intuitively Correct Interaction (5)**
    1. Intuitive perceptibility of functions and interaction elements (USA_7)
    2. Effort required to learn the operation is low (USA_16)
    3. Operation is easy to memorise (USA_17)
    4. Accidental activation of the system does not lead to hazards (DAM_7)
    5. Accidental movements of people (e.g., slipping, tripping, falling) do not lead to hazards (DAM_10)

2. **Comprehensibility of Form and Design (9)**
    1. Intuitive understandability of functions and interaction elements (USA_8)
    2. Each interaction element is unique in terms of its use (USA_15)
    3. Operation is internally consistent, i.e., design for elements with similar function is also similar (USA_19)
    4. Operation is familiar from other areas to users (USA_20)
    5. Design fulfils expected fundamental functions (UEX_4)
    6. The system is simple and small, redundant functions are reduced as much as possible (UEX_8)
    7. Form and shape complexity, i.e., the total number of lines and curves, the change in their direction, and the amount and type of combinations of shapes that occur, is small (UEX_10)
    8. Interaction elements can be detected by means of two sensory capabilities (ACC_1)
    9. Interaction elements are intuitively recognisable (ACC_3)

3. **Comprehensibility of System Interaction and System Output (5)**
    1. Interaction elements are easily reachable when needed (USA_5)
    2. Direct feedback about system status and operations (USA_6)
    3. Users do not get the feeling of incompetence in operation (UEX_1)
    4. Product form fits the context of use (UEX_6)

    5.    System uses a language that is easy to understand (ACC_8)

**4.**    Error and Hazard Avoidance in Operation (7)

    1.    Helpful information is available when needed (USA_9)
    2.    Users can correct errors when they occur (USA_13)
    3.    Elements and functions are clearly and noticeably labelled (ACC_4)
    4.    Moving parts do not lead to mechanical hazards (DAM_3)
    5.    Moving masses do not lead to mechanical hazards (DAM_4)
    6.    Elastic elements do not lead to mechanical hazards (DAM_5)
    7.    Movement of people around the facility does not lead to hazards (DAM_12)

**5.**    Effective and Efficient Interaction (14)

    1.    Interaction elements are simple and clear (USA_1)
    2.    Minimal number of actions required to complete task (USA_2)
    3.    Operations are effective, i.e., require low effort (USA_3)
    4.    Operations are efficient, i.e., require little time (USA_4)
    5.    Design prevents errors by users to a large extent (USA_14)
    6.    Operation can be adapted to the needs of the user (USA_18)
    7.    Continuous work is possible/no forced interruptions (UEX_2)
    8.    Design has recognisable technological advance (UEX_5)
    9.    Total number of visible individual parts is rather small (UEX_7)
    10.    Elements and functions are visually merged or organised into sets (UEX_9)
    11.    Product form follows a cohesive style characterised by unity, contrast, or balance (UEX_12)
    12.    Operating target can be reached via two independent paths (ACC_2)
    13.    Working speed is reasonable and/or can be adjusted to the user (ACC_7)
    14.    Insufficient information does not lead to hazards (DAM_11)

**6.**    Inherently Safe and Ergonomic Operation (14)

    1.    Users have control over all functions (USA_10)
    2.    Users have the feeling of direct manipulation (USA_11)
    3.    Functions are easy to control (USA_12)
    4.    Operation and interaction elements are predictable for the user (USA_21)
    5.    Operation creates a feeling of security/prevents feeling of insecurity (UEX_3)
    6.    Proportions follow a harmonic design; dimensions differ for purely ergonomic and functional aspects (UEX_11)
    7.    Interaction elements are accessible with reduced mobility (ACC_5)
    8.    Interaction elements can be operated with little physical strength (ACC_6)
    9.    The shape does not lead to mechanical hazards (DAM_1)
    10.    Protruding elements do not lead to mechanical hazards (DAM_2)
    11.    Unhealthy postures are avoided (DAM_6)
    12.    Failure of the system does not lead to hazards (DAM_8)
    13.    There is no danger to people if safety devices are bypassed (DAM_9)
    14.    Emergency stop function is provided/absence of emergency stop does not lead to hazards (DAM_13)

## 4. Discussion

The evaluation with the HCM shows patterns and trends with regard to better human-centredness when using additional methods that can be assigned to the area of human-centredness in the broadest sense, in contrast to not using any additional method. This also corresponds to our definition from Figure 1 that activities that take place without direct user interaction but provide information about the user or the context of use are also to be defined as human-centred, since they are capable of improving human-centred design quality goals in the design. Due to the lack of significance of the results, however, this answer cannot be given conclusively. The results seem to be independent of other influencing factors examined, such as the choice of product examples, the order in which

the experiments were carried out, and the abilities of the students, as we deliberately mixed these factors within the design of the experiment, although it is also difficult to give definitive answers here. However, other measurement methods that were used comparably early in the design process, as in Genco et al. [25], turned out to be quite applicable and usable too. We therefore assume that the weakness lies more in the experimental design than in the basic idea of the metric. However, this was primarily designed to test the internal consistency and suitability of the HCM for an appropriate task.

Apart from the limitations already mentioned and partly addressed in the experimental design, the early stage of development of the concepts is a further one. Thus, only those functions could be evaluated by the metric that were also expressed in concrete terms by the participants. In the case of the vacuum cleaner, for example, one student described the changing of brush attachments, but the description of other basic operating processes, i.e., switching on/off, adjusting power, etc., was largely neglected; these could not be evaluated or could only be anticipated to a limited extent from the requirement and function lists. Nevertheless, even in the case of missing or insufficient descriptions, decisions were usually made in favour of the concepts. It is therefore possible that with increasing detail and repeated application of the evaluation at a later point in time, the scores in the HCM will decrease and the concept may be found to be less user-friendly or less accessible than previously assumed. Accordingly, it is not possible to choose the most user-friendly, most accessible, etc., concept with high probability at this very early development stage.

There are also weaknesses in the metric in terms of its applicability and meaningfulness, as can be seen from the overview of items that were partially not applicable (Figure 7), as well as the fact that some items were always fulfilled by all concepts. On the one hand, the items on visual aesthetics in particular reach their limits when they are applied to very early product development phases and thus very rough product concepts. Furthermore, the items on the avoidance of use-related damage show obvious weaknesses in their applicability to the example of household appliances. After all, these items were primarily derived from the Machinery Directive. Larger masses or forces from which a hazard could emerge, however, are generally not to be feared in the case of household appliances. The extent to which this is a problem for the meaningfulness of the metric as such remains open.

In the context of the product examples used, there is another possible limitation for the HCM. Within the study, it was only applied to physical everyday devices and not to digital user interfaces. Due to the form of the HCM and the topics that it addresses, we assume reduced applicability or a necessity for adaptation for application to purely digital systems. For example, all items that stem from the prevention of use-related damage, as well as some others, are very strongly oriented towards physical devices and can only be logically applied to them.

## 5. Conclusions

A research field as mixed as human-centred design needs applicable and reliable definitions and techniques to provide the basis for the comparability of its associated methods, on the one hand, and to estimate the fulfilment of human-centred design goals in product design at an early stage on the other hand. In this paper, we addressed the following research question: How can the human-centred idea be measured and evaluated based on resulting product concepts? As a first step and initial study to answer this question, we designed an empirical study applying the recommendations from [59,60] and following the design method validation models of [57,58], which primarily aimed to test the internal consistency, completeness, and applicability of the metric (HCM) developed to answer the research question. Regarding internal consistency, which, in the case of a metric, is a statement about how well the individual items measure the construct under investigation and no other, we were able to calculate a good Cronbach's alpha value, i.e., none of the items showed evidence of measuring a construct other than human-centredness. Regarding completeness, based on the items that showed no change due to variation in the independent variable and were marked as likely to be fulfilled for (almost)

all concepts, there is a need to improve the formulations so that they better measure the addressed variables. In terms of applicability, we found limitations in the roughness of the initial design, especially for usability and user experience items. More detailed concepts or user descriptions would be necessary here. It is unclear to what extent the reformulation of the items could compensate for this limitation. As a second objective, this study also aimed to examine the extent to which the HCM is actually suitable for demonstrating the influence of an additional human-centred activity on the product design and thus examine the HCM's functionality. While the descriptive statistics and the graphical analysis showed certain tendencies suggesting improved human-centredness through the use of additional methods, the statistical analysis showed no significance in the results. The hypotheses about the impact of these methods could neither be conclusively confirmed nor rejected.

Based on the success of human-centred methods in general, especially those with active user involvement, we assume that these methods fundamentally and undoubtedly have added value for the quality of human–system interaction and the user experience of products as a whole. This shall not be questioned here. However, we could not clearly prove or measure the strength of this effect for the three methods examined in this study. Only the potential of the HCM in evaluating human-centredness was presented and provided insights into the challenges and intricacies of measuring the strength of human-centredness in the early stages of product development. This study contributes to the ongoing discourse on the role of human-centred design in product development and highlights the importance of further research into effective methods of assessing and quantifying human-centredness. The questions that continue to await answers are therefore the following: How much user interaction is necessary in different development processes, or how much user-oriented work is too little to achieve added value for product design? How early can this effect be measured in the quality of the product concepts, or how early in the development process is too early for evidence?

This contribution can be summarised as an initial study on the structure and applicability of a new metric for human-centredness evaluation. Further studies for the validation of the metric are thus needed. Future work should focus on applications with more concepts and fewer variables to ensure the increased internal validity of the study design. Future studies should therefore focus on the evaluation of the functionality and usefulness of the HCM. Conceivably, only one human-centred design method could be chosen and the effects measured with and without its use. In this context, the HCM should be also further improved and optimised. The items should be examined more closely for overlapping meanings and for imprecise formulations, and the applicability for other users should be checked in order to further address the user-friendliness of the HCM.

**Author Contributions:** Conceptualisation, O.S. and D.K.; methodology, O.S.; validation, O.S. and D.K.; formal analysis, O.S.; investigation, O.S.; resources, O.S.; data curation, O.S.; writing—original draft preparation, O.S.; writing—review and editing, D.K.; visualisation, O.S.; supervision, D.K.; project administration, O.S. and D.K. All authors have read and agreed to the published version of the manuscript.

**Funding:** Publishing fees supported by Funding Programme Open Access Publishing of Hamburg University of Technology (TUHH).

**Institutional Review Board Statement:** Not applicable.

**Informed Consent Statement:** Informed consent was obtained from all participants involved in the study to process their data.

**Data Availability Statement:** The data presented in this study are available on request from the corresponding author. The data are not publicly available due to privacy reasons.

**Conflicts of Interest:** The authors declare no conflict of interest.

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
