# Peer review of "The Human-Centredness Metric: Early Assessment of the Quality of Human-Centred Design Activities"

_applsci, doi:10.3390/app132112090_

Round 1
Reviewer 1 Report
Comments and Suggestions for Authors
The paper covers an interesting topic of how to evaluate UCD activities, which is an important topic.
The introduction to the problem is well written. I am in the HCI field, were Human-centred design is one of the central processes for developing software with good user experience. More references to the HCI field could be inserted here, since there are some studies, comparing the usage of various Human-centred design activities that are not mentioned in the introduction, but other than that that section reads well.
The problem I have with the paper is that there are 4 participants in the study, all university students and each of them use one method and I think it is vague to base a metric on a study with only 4 participants. The human-centred methods chosen are universal design princliples, which is not a method per se, and or an activity, it is a set of principles. The process of using the principles is an activity. This also relates to the third method the dialogue principles. The persona method is a human-centred activity.
I find it strange to choose two out of three activities for evaluating the metrics, that are not really human-centred.
More detailed comments:
1. On figure 2, Interviews are not typically grouped as evaluation methods.
2. In section 2, it says that approach wherein any endeavour involving the gathering of data or information directly or indirectly from, with, or by the user about the context of use as part of a product development process is classified as adhering to the principles of human-centred design. Where does this come from? Why they emphasis on context of use? I think this needs better clarification.
3. In line 169 on pg. 5, there is the argument that HCM should be expert-nased and not involve users. I think the argumentation is unclear. I have a hard time following the rational.
Comments on the Quality of English Language
The quality of the English language is mostly good.
Author Response
Thank you for your feedback.
Please see attachment.

Reviewer 2 Report
Comments and Suggestions for Authors
This manuscript is present and apply a new metric - the Human-Centredness Metric (HCM). This manuscript aim to closely follow existing standards and develop a metric for early assessment of product concepts based on the human-centred quality goals defined. The application of human-centred design methods will lead to a significant change in the human-centred quality goals of the developed concepts. We agree with the hypothesis of the most basic logical aspect of the manuscript. Moreover, the data analysis in this manuscript is reasonable and the language description is appropriate. The logic is relatively clear. I agree with the conclusion in the manuscript.
Author Response
Thank you.
No action necessary.
Reviewer 3 Report
Comments and Suggestions for Authors
1) I guess you want to say: "human-centeredness" is gaining increasing importance (row 29).
2) The statement in lines 53-57 requires citation or at least a broader justification of the authors: "Despite the international standards, their definitions and guidelines, the research field of human-centred design lacks conceptual foundations for methodological research. For instance, the definition of a user-centred or human-centred activity remains ambiguous. Does this only pertain to activities that have been done with the involvement of end users, are all activities leading to a better human-system interaction human-centered, or what exactly are the boundaries of human-centred approaches?".
3) The statement in lines 91-93 requires additional explanations or citations: "It therefore remains unclear what constitutes human-centred design methods, whether they have a direct effect on the conception of products and whether this effect can be measured."
4) Explain in more detail (possibly with citations) how you chose the objective of your article: "How can human-centredness be measured and evaluated based on resulting product concepts?".
5) Modify lines 113-114, with discussions in section 4 and conclusions in section 5.
6) Figure 1 requires additional presentation.
7) Figure 2 requires additional discussion.
8) Figure 3 requires additional discussion.
9) Please explain more clearly how you ensure the repeatability of the results obtained experimentally.
10) Unfortunately, Figure 4 requires a higher resolution.
11) Figures 5 and 6 require additional discussions.
12) Please discuss the results obtained in Tables 1 and 2.
13) Try to explain in more detail the algorithms used.
14) Try in section 4 to highlight again all the main contributions of the work, and possibly the future directions and perspectives offered by them.
Moderate editing of English language required
Author Response
Thank you for your feedback.
Please see the attachment.

Round 2
Reviewer 3 Report
Comments and Suggestions for Authors
Accept in present form
Comments on the Quality of English LanguageMinor editing of the English language is required